# Benefits of Overparameterization in Single-Layer Latent Variable Generative Models

## Abstract

One of the most surprising and exciting discoveries in supervising learning was the benefit of overparameterization (i.e. training a very large model) to improving the optimization landscape of a problem, with minimal effect on statistical performance (i.e. generalization). In contrast, unsupervised settings have been under-explored, despite the fact that it has been observed that overparameterization can be helpful as early as Dasgupta & Schulman (2007). In this paper, we perform an exhaustive study of different aspects of overparameterization in unsupervised learning via synthetic and semi-synthetic experiments. We discuss benefits to different metrics of success (recovering the parameters of the ground-truth model, held-out log-likelihood), sensitivity to variations of the training algorithm, and behavior as the amount of overparameterization increases. We find that, when learning using methods such as variational inference, larger models can significantly increase the number of ground truth latent variables recovered.

## 1 INTRODUCTION

Unsupervised learning is an area of intense focus in recent years. In the absence of labels, the goal of unsupervised learning can vary. Generative adversarial networks, for example, have shown promise for density estimation and synthetic data generation. In commercial applications, unsupervised learning is often used to extract features of the data that are useful in downstream tasks. In the sciences, the goal is frequently to confirm or reject whether a particular model fits well or to identify salient aspects of the data that give insight into underlying causes and factors of variation.

Many unsupervised learning problems of interest, cast as finding a maximum likelihood model, are computationally intractable in the worst case. Though much theoretical work has been done on provable algorithms using the method of moments and tensor decomposition techniques (Anandkumar et al., 2014; Arora et al., 2017; Halpern & Sontag, 2013), iterative techniques such as variational inference are still widely preferred. In particular, variational approximations using recognition networks have become increasingly popular, especially in the context of variational autoencoders (Mnih & Gregor, 2014). Intriguingly, it has been observed, e.g. by Yeung et al. (2017), that in practice many of the latent variables have low activations and hence not used by the model.

Related phenomena have long been known in supervised learning: it was folklore knowledge among practitioners for some years that training larger neural networks can aid optimization, yet not affect generalization substantially. The seminal paper by Zhang et al. (2016) thoroughly studied this phenomenon with synthetic and real-life experiments in the supervised setting. In brief, they showed that some neural network architectures that demonstrate strong performance on benchmark datasets are so massively overparameterized that they can "memorize" large image data sets (they can perfectly fit a completely random data set of the same size). Subsequent theoretical work provided mathematical explanations of some of these phenomena (Allen-Zhu et al., 2018; Allen-Zhu & Li, 2019). In contrast, overparameterization in the unsupervised case has received much less attention.

This paper aims to be a controlled empirical study that measures and disentangles the benefits of overparameterization in unsupervised learning settings. More precisely, we consider the task of fitting two common latent-variable models – a discrete factor analysis model using a noisy-OR parameterization and sparse coding (interpreted as a probabilistic model, see Section 2). Through experiments on synthetic and semi-synthetic data sets, we study the following aspects:

- **Latent variable recovery**: We show that larger models increase the number of ground truth latent variables recovered, as well as the number of runs in which the all ground truth latent variables are recovered. Furthermore, we show that recovering the ground-truth latent variables from the overparameterized solutions can be done via a simple filtering step: the optimization tends to converge to a solution in which all latent variables that do not match ground truth latent variables can either be discarded (i.e. have low prior probability) or are near-duplicates of other matched latent variables.

- **Effects of extreme overparameterization:** We show that while the benefits of adding new latent variables have diminishing returns, the harmful effects of extreme overparameterization are at most minor. Both the number of ground truth recoveries and the held-out log-likelihood do not worsen significantly as the number of latent variables increases. Instead, sometimes the performance continues to increase even with 10 times the true number of latent variables.

- **Effects of training algorithm**: We show that changes to the training algorithm, such as significantly increasing the batch size or using a different variational posterior, do not affect the beneficial effects of overparameterization significantly. For learning noisy-OR networks, we test two algorithms based on variational learning: one with a logistic regression recognition network, and one with a mean-field posterior (e.g. see Wainwright et al. (2008)).

- **Latent variable stability over the course of training**: One possible explanation for why overparameterization helps is that having more latent variables increases the chances that at least one initialization will be close to each of the ground-truth latent variables. (This is indeed the idea of Dasgupta & Schulman (2007)). This does not appear to be the dominant factor here. We track the "matching" of the trained latent variables to the ground truth latent variables (matching = minimum cost bipartite matching, with a cost based on parameter closeness), and show that this matching changes until relatively late in the training process. This suggests that the benefit of overparameterization being observed is not simply due to increased likelihood of initializations close to the ground truth values.

In this investigation, we use simple generative models so that we can exert maximal control over our experiments. With more complex, deep generative models, there is a lot of latitude in choosing the architecture of the model when overparameterizing, making it more difficult to disentangle the importance of the choice of architecture from overparameterization in general.

## 2 LEARNING OVERPARAMETERIZED GENERATIVE MODELS

We focus on the task of fitting two commonly used latent-variable models: noisy-OR networks and sparse coding. Beyond the ubiquity of these models, the reason we chose them is that they provide the "simplest" architectures for a generative model: a single latent layer, albeit with different activation functions. This makes the choice of "overparameterized" architecture non-controversial: we simply include more latent variables.

**Noisy-OR Networks:**  A *noisy-OR network* (Pearl, 1988) is a bipartite directed graphical model, in which one layer contains binary latent variables and the other layer contains binary observed variables. Edges are directed from latent variables to observed variables. The model has as parameters a set of prior probabilities $\pi \in [0,1]^m$ for the latent variables, a set of noise probabilities $l \in [0,1]^n$ for the observed variables, and a set of weights $W \in \mathbb{R}_+^{n \times m}$ for the graph. If the latent variables are denoted as $h \in \{0,1\}^m$ and the observed variables as $x \in \{0,1\}^n$, the joint probability distribution specified by the model factorizes as: $p(x,h) = p(h) \prod_{j=1}^{n} p(x_j|h)$, where

$$p(h) = \prod_{i=1}^{m} \pi_i^{h_i} (1-\pi_i)^{1-h_i}, \qquad p(x_j = 0|h) = (1-l_i) \prod_{i=1}^{m} \exp(-W_{ji} h_i).$$

It is common to refer to $\exp(-W_{ji})$ as the failure probability between $h_i$ and $x_j$ (i.e. the probability with which, if $h_i = 1$, it "fails to activate" $x_j$).

*Training algorithm:* We optimize an approximation of the likelihood of the data under the model, the evidence lower bound (ELBO). This is necessary because direct maximum likelihood optimization is intractable. If the joint pdf is $p(x, h; \theta)$, we have, using the notation in Mnih & Gregor (2014):

$$\log p(x; \theta) \geq \mathbb{E}_{q(\cdot|x;\phi)} [\log p(x,h;\theta) - \log q(h|x;\phi)] = \mathcal{L}(x, \theta, \phi)$$

where $q(\cdot|x;\phi)$ is a variational posterior, also known as a recognition network in this setting. When $q(h|x;\phi) = p(h|x;\theta), \forall h$, the inequality becomes equality; however, it is intractable to compute $p(h|x;\theta)$. Instead, we will assume that $q$ belongs to a simpler family of distributions: a logistic regression network parameterized by weights $w_i \in \mathbb{R}^n$ and bias terms $b_i \in \mathbb{R}, \forall i \in \{1, ..., m\}$, s.t.:

$$q(h|x;\phi) = \prod_{i=1}^{m} \sigma(x \cdot w_i + b_i)^{h_i}(1 - \sigma(x \cdot w_i + b_i))^{1-h_i}.$$

Then, we maximize the lower bound by taking gradient steps w.r.t. $\theta$ and $\phi$. Furthermore, to improve the estimation of the gradients, we use variance normalization and input-dependent signal centering, as in Mnih & Gregor (2014). For the input-dependent signal centering, we use a two-layer neural network with $100$ hidden nodes in the second layer and $\tanh$ activation functions.

*Extracting the ground-truth latent variables:* As we are training an overparameterized model, we need to filter the learned latent variables to extract latent variables corresponding to the ground-truth variables. First, we discard all latent variables that are *discardable*, namely have learned prior probability less than $0.02$ or have all learned failure probabilities of the related observable variables greater than $0.8$. Then, for every pair of latent variables that are *duplicates* (measured as having failure probability vectors closer than $4.0$ in $l_1$ distance), we discard the one with lower prior probability – such that for any cluster of duplicate latent variables, only the one with the largest prior probability survives.

**Sparse Coding:** A *sparse coding* model is specified by a matrix $A \in \mathbb{R}^{n \times m}$ with $||A_i||_2 = 1, \forall i$ (i.e. unit columns). Samples are generated from this model according to $x = Ah$, with $h \in \mathbb{R}^m$, $||h||_1 = 1$ and $||h||_0 = k$ (i.e. sparsity $k$). The coordinates of the vector $h$ play the role of the latent variables, and the distribution $h$ is generated from is as follows: first, uniformly randomly choose $k$ coordinates of $h$ to be non-zero; next, sample the values for the non-zero coordinates uniformly in $[0, 1]$; finally, renormalize the non-zero coordinates so they sum to $1$.

*Training algorithm:* We use a simple alternating-minimization algorithm given in Li et al. (2016). It starts with a random initialization of $A$, such that $A$ has unit columns. Then, at each iteration, it "decodes" the latent variables $h$ for a batch of samples, s.t. a sample $x$ is decoded as $h = \max(0, A^\dagger x - \alpha)$, for some fixed $\alpha$ and the current version of $A$. After decoding, it takes a gradient step toward minimizing the "reconstruction error" $||Ah - x||_2^2$, and then re-normalizes the columns of $A$ such that it has unit columns. Here, overparameterization means learning a matrix $A \in \mathbb{R}^{n \times s}$ with $s > m$, where $m$ is the number of columns of the ground truth matrix.

*Extracting the latent variables:* Similarly as in the noisy-OR case, we are training an overparameterized model, so to extract latent variables which correspond to the ground-truth variables we need to a filtering step. First, we apply the decoding step $h = \max(0, A^\dagger x - 0.005)$ to all samples $x$ in the training set, and mark as "present" all coordinates in the support of $h$. Second, we discard the columns that were never marked as "present". The intuition is rather simple: the first step is a proxy for the prior in the noisy-OR case (it captures how often a latent variable is "used"). The second step removes the unused latent variables. (Note, one can imagine a softer removal, where one removes the variables used less than some threshold, but this simpler step ends up being sufficient for us.)

## 3 EMPIRICAL STUDY FOR NOISY-OR NETWORKS

We study the effect of overparameterization in noisy-OR networks using $7$ synthetic data sets:

**(1)** The first, IMG, is based on Šingliar & Hauskrecht (2006). There are $8$ latent variables and $64$ observed variables. The observed variables represent the pixels of an $8 \times 8$ image. Thus, the connections of a latent variable to observed variables can be represented as an $8 \times 8$ image (see Figure 1). Latent variables have priors $\pi_i = 0.25$. All failure probabilities different from $1.0$ are $0.1$.

**(2)** The second, PLNT, is semi-synthetic: we learn a noisy-OR model from a real-world data set, then sample from the learned model. We learn the model from the UCI plants data set (Lichman et al., 2013), where each data point represents a plant that grows in North America and the 70 binary features indicate in which states and territories of North America it is found. The data set contains 34,781 data points. The resulting noisy-OR model has $8$ latent variables, prior probabilities between $0.05$ and $0.20$, and failure probabilities either less than $0.5$ or equal to $1$.

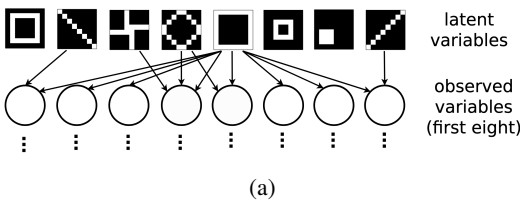 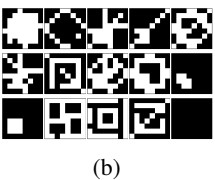

(a)                            (b)

Figure 1: (a) Configuration of the IMG noisy-OR model. In the first row, each $8 \times 8$ image represents a latent variable. Each pixel in an $8 \times 8$ image represents the failure probability of the latent variable with the corresponding observed variable (white pixels correspond to failure probabilities different from $1.0$). In the second row, each node represents an observed variable; the observed variables corresponding to the first row of the $8 \times 8$ images are shown. The edges show failure probabilities different from $1.0$. (b) Samples of the IMG data set. Each $8 \times 8$ image represents a sample, and each pixel represents an observed variable (white pixels correspond to $1$).

The next three data sets are based on randomly generated models with $8$ latent and $64$ observed variables (same as IMG):

- **(3)** UNIF: Each latent variable's prior is sampled $\pi_i \sim \mathcal{U}[0.1, 0.3]$ and it is connected to each observation with probability $0.25$. If connected, the corresponding failure probability is drawn from $\mathcal{U}[0.05, 0.2]$; otherwise it is $1.0$.

- **(4)** CON8: $\pi_i = 1/8$ for all $i$. Each latent variable is connected to exactly $8$ observed variables, selected at random. If connected, the failure probability is $0.1$; otherwise it is $1.0$.

- **(5)** CON24: Same as CON8, but each latent variable is connected to $24$ observed variables.

The rationale for the previous two distributions is to test different densities for the connection patterns.

The final two are intended to assess whether overparameterization continues to be beneficial in the presence of model misspecification, i.e. when the generated data does not truly come from a noisy-OR model, or when there are additional (distractor) latent variables that occur with low probability.

**(6)** IMG-FLIP: First, generate a sample from the IMG model described above. Then, with probability $10\%$, flip the value of every fourth observed variable in the sample (i.e. $x_0$, $x_4$, ...).

**(7)** IMG-UNIF: This model has $16$ latent variables and $64$ observed variables. The first $8$ latent variables are those of the IMG model, again with prior probability $0.25$. We then introduce $8$ more latent variables from the UNIF model, with prior probabilities $0.05$ each.

For all models except PLNT, noise probabilities are set to $0.001$. PLNT uses the learned noise probabilities. To create each data set, we generate $10,000$ samples from the corresponding model. We split these samples into a training set of $9,000$ samples and a validation set of $1,000$ samples. Samples are generated exactly once from the ground truth model and re-used in all experiments with that model. For the randomly generated models, we generate the ground truth model exactly once.

To count how many ground truth latent variables are recovered, we perform minimum cost bipartite matching between the ground truth latent variables and the recovered latent variables. The cost of matching two latent variables is the $l_\infty$ distance between their weight vectors (removing first the variables with prior probability lower than $0.02$). After finding the optimal matching, we consider as recovered all ground truth latent variables for which the matching cost is less than $1.0$. Note the algorithm may recover the ground truth latent variables without converging to a maximum likelihood solution because we do not require the prior probabilities of the latent variables to match (some of the latent variables may be split into duplicates) and because the matching algorithm ignores the state of the unmatched latent variables. We discuss this in more detail in Section 3.

**Overparameterization Improves Ground Truth Recovery and Log-likelihood:** For all data sets, we test the recognition network algorithm using $8$ latent variables (i.e. no overparameterization), $16$, $32$, $64$, and $128$. For each experiment configuration, we run the algorithm $500$ times with different random initializations of the generative model parameters. We report in Figure 2 the average number of ground truth latent variables recovered and the percentage of runs with full ground truth recovery (i.e. where $8$ ground truth latent variables are recovered). We see that in all data sets,

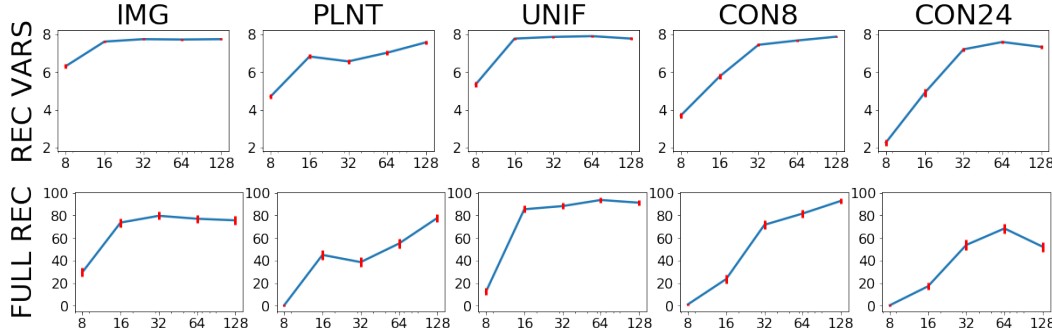

Figure 2: Performance of the noisy-OR network learning algorithm. The plots show statistics for 500 runs of the algorithm with random initializations on different data sets with different number of latent variables. The $y$-axis on the top row denotes the average number of ground truth latent variables recovered and in the bottom the percentage of runs with full ground truth recovery. The 95% confidence intervals are shown in red bars.

overparameterization leads to significantly improved metrics compared to using 8 latent variables. The same trends are observed for the held-out log-likelihood. See the Appendix (E, Table 1) for more detailed numerical results for these experiments.

**Harm of Extreme Overparameterization Is Minor; Benefits Are Often Significant:** The results suggest that there may exist an optimal level of overparameterization for each data set, after which overparameterization stops conferring benefits (the harmful effect then may appear because larger models are more difficult to train). The peak happens at 32 latent variables for IMG, at 128 for PLNT, at 64 for UNIF, at 128 for CON8, and at 64 for CON24. Therefore, overparameterization can continue to confer benefits up to very large levels of overparameterization. In addition, even when 128 latent variables are harmful with respect to lower levels of overparameterization, 128 latent variables lead to significantly improved metrics compared to no overparameterization.

**Unmatched Latent Variables Are Discarded or Duplicates:** When the full ground truth is recovered in an overparameterized setting, the unmatched latent variables usually fall into two categories: discardable or duplicates, as described in Section 2. To test this observation systematically, we use the filtering step described in that section.

We applied the filtering step to all experiments reported in Figure 2, in the runs where the algorithm recovered the full ground truth. In nearly all of these runs, the filtering step keeps exactly 8 latent variables that match the ground truth latent variables. Exceptions are 255 runs out of a total of 6929 runs (i.e. 3.68%); in these exception cases, the filtering step tends to keep more latent variables, out of which 8 match the ground truth, and the others have higher failure probabilities (but nonetheless lower than the threshold of 0.80). Note that the solutions containing duplicates are not in general equivalent to the ground truth solutions in terms of likelihood: we give an illustrative example in Appendix (H).

**Batch Size Does Not Change the Effect:** We also test the algorithm using batch size 1000 instead of 20. Although the performance decreases – as may be expected given folklore wisdom that stochasticity is helpful in avoiding local minima – overparameterization remains beneficial across all metrics. This shows that the effect of overparameterization is not tied to the stochasticity conferred by a small batch size. For example, on the IMG data set, we recover on average 5.91, 7.17, and 7.32 ground truth latent variables for learning with 8, 16, and 32 latent variables, respectively. See the Appendix (E, Table 2) for detailed results.

**Variational Distribution Does Not Change the Effect:** To test the effect of the choice of variational distribution, on all data sets, we additionally test the algorithm using a per-sample mean-field variational posterior instead of the logistic regression recognition network. The variational posterior models the latent variables as independent Bernoulli. In each epoch, *for each sample*, the variational posterior is updated from scratch until convergence using coordinate ascent, and then a gradient update is taken w.r.t. the parameters of the generative model. Thus, this approximation is more flexible than the recognition network: the Bernoulli parameters can be learned separately for each sample instead of being outputs of a global logistic regression network.

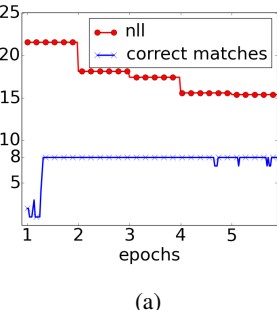

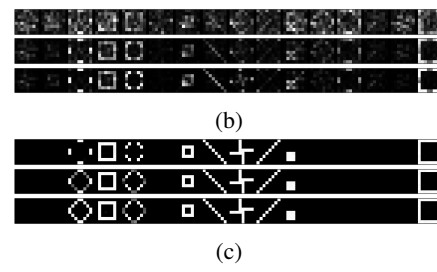

(b)

(a)

(c)

Figure 3: State of the optimization process on a successful run of the noisy-OR network learning algorithm on the IMG data set with 16 latent variables. (a) The blue line (with "x") shows the number of latent variables matched to the same ground truth latent variable as at the end of the optimization. The red line (with "o") is the negative held-out log-likelihood. The graph is truncated at 5 epochs. (b) The shapes of the latent variables after $1/9$ epochs, $2/9$ epochs, and $3/9$ epochs. (c) The shape of the latent variables after 10 epochs, 20 epochs, and 30 epochs.

Though the specific performance achieved on each data set often differs significantly from the previous results, overparameterization still leads to clearly improved metrics on all data sets. For example, on the IMG data set, we recover on average 6.72, 7.52, and 7.23 ground truth latent variables for learning with 8, 16, and 32 latent variables, respectively. See the Appendix (E, Table 3).

**Matching to Ground-truth Latent Variables Is Unstable:** To understand the optimization dynamics better, we inspect how early the recovered latent variables start to converge toward the ground truth latent variables they match in the end. If this convergence started very early, it could indicate that each latent variable converges to the closest ground truth latent variable – then, overparameterization would simply make it more likely that each ground truth latent variable has a latent variable close to it at initialization.

The story is more complex. First, early on (especially within the first epoch), there is significant conflict between the latent variables, and it is difficult to predict which ground truth latent variable they will converge to. We illustrate this in Figure 3 for a run that recovers the full ground truth on the IMG data set when learning with 16 latent variables. In part (a) of the Figure, at regular intervals in the optimization process, we matched the latent variable to the ground truth latent variables and counted how many pairs are the same as at the end of the optimization process. Especially within the first epoch, the number of such pairs is small, suggesting the latent variables are not "locked" to their final state. Part (b) of the Figure pictorially depicts different stages in the first epoch of the run with 16 latent variables, clearly showing that in the beginning there are many latent variables that are in "conflict" for the same ground truth latent variables.

Second, even in the later stages of the algorithm, it is often the case that the contribution of one ground truth latent variable is split between multiple recovered latent variables, or that the contribution of multiple ground truth latent variables is merged into one recovered latent variable. This is illustrated in part (c) (the same successful run depicted in Figure 3 (b)), which shows multiple later stages of the optimization process which contain "conflict" between latent variables. See the Appendix (F) for the evolution of the optimization process across more intervals.

Of course, the observations above do not rule out the possibility that closeness at initialization between the latent variables and the ground truth latent variables is an important ingredient of the beneficial effect. However, we showed that the optimization process remains complex even in cases with overparameterization.

**Effects of Model Mismatch:** In the experiments with model mismatch, we still use the noisy-OR network learning algorithm from Section 3. We show that, in both data sets, overparameterization allows the algorithm to recover the underlying IMG latent variables more accurately, while modeling the noise with extra latent variables. In general, we think that in misspecified settings the algorithm tends to learn a projection of the ground truth model onto the specified noisy-OR model family, and that overparameterization often allows more of the noise to be "explained away" through latent variables.

For both data sets, the first bump in the recovery metrics happens when we learn with 9 latent variables, which allows 8 latent variables to be used for the IMG data set, and the extra latent variable to capture some of the noise. After that, more overparameterization increases the accuracy even further. For example, on the IMG-FLIP data set, we recover on average 4.40, 5.59, and 5.99 ground truth latent variables for learning with 8, 9, and 10 latent variables, respectively. See the Appendix (E, Table 4) for detailed results. Also, see the Appendix (G) for examples of the latent variables recovered in successful runs.

For IMG-FLIP, in successful runs, the algorithm tends to learn a model with an extra latent variable with significant non-zero prior probability that approximates the shape of the noise (i.e. a latent variable with connections to every fourth observed variable). For IMG-UNIF, the algorithm uses the extra latent variables to capture the tail latent variables. In this case, the algorithm often merges many of the tail latent variables, which is likely due to their smaller prior probabilities.

## 4  EMPIRICAL STUDY FOR SPARSE CODING

We find that the conclusions for sparse coding are qualitatively the same as for the noisy-OR models. Thus, for reasons of space, we only describe them briefly; see Appendix (E, Table 5) for full details.

We again evaluate using synthetic data sets. $A$ is sampled in pairs of columns such that the angle between each pair is a fixed $\gamma$. Specifically, we first generate 12 random unit columns; then, for each of these columns, we generate another column that is a random rotation at angle $\gamma$ of the original column. As a result, columns in different pairs have with high probability an absolute value inner product of approximately $\frac{1}{\sqrt{n}}$ (i.e., roughly orthogonal), whereas the columns in the same pair have an inner product determined by the angle $\gamma$. The smaller the angle $\gamma$, the more difficult it is to learn the ground truth, because it is more difficult to distinguish columns in the same pair. We construct two data sets, the first with $\gamma = 5°$ and the second with $\gamma = 10°$.

We experimented with learning using a matrix with 24 columns (the true number) and with 48 columns (overparameterized). To measure how many ground truth columns are recovered, we perform minimum cost bipartite matching between the recovered columns and the ground truth columns. As cost, we use the $l_2$ distance between the columns and consider correct all matches with cost below 0.002. We measure the error between the recovered matrix and the ground truth matrix as the sum of costs in the bipartite matching result (including costs larger than the threshold).

As in the case of noisy-OR networks, overparameterization consistently improves the number of ground truth columns recovered, the number of runs with full ground truth recovery, and the error. For example, for $\gamma = 10°$, learning with 24 columns gives on average 19.77 recovered columns, 17.2% full recoveries, and 2.28 error, while learning with 48 columns gives 23.84 recovered columns, 88.4% full recoveries, and 0.04 average error.

## 5  RELATED WORK

On the empirical side, a few previous papers have considered overparameterized models for unsupervised learning and evaluated using synthetic distributions where assessing ground truth recovery is possible. Šingliar & Hauskrecht (2006) and Dikmen & Févotte (2011) observe that, in overparameterized settings, the unnecessary latent variables are discarded and that the log-likelihood does not decrease. However, they do not observe any beneficial effects. In the former case, this is likely because their variational approximation is too weak; in the latter case, it is because the ground truth is already recovered without overparameterization. Hughes et al. (2015) show that larger levels of overparameterization can lead to better held-out log-likelihood than lower levels of overparameterization for some learning algorithms. Separately, they show that some of their overparameterized learning algorithms recover the ground truth; however, they do not investigate how this ability varies as a function of the amount of overparameterization.

On the theoretical side, to our knowledge, the earliest paper that points out a (simple) benefit of overparameterization is Dasgupta & Schulman (2007) in the context of recovering the means of $k$ well-separated spherical Gaussians given samples from the mixture. They point out that using $O(k \ln k)$ input points as "guesses" (i.e. initializations) for the means allows us to guarantee, by the

coupon collector phenomenon, that we include at least one point from each component in the mixture – which would not be so if we only used $k$. A filtering step subsequently allows them to recover the $k$ components. We could reasonably conjecture that the benefits in our setting are due to a similar reason: overparameterization could guarantee at least one trained latent variable close to each ground truth latent variable. However, simple high-dimensional concentration of measure easily implies that the probability of having a reasonably highly-correlated initialization (e.g. in terms of inner product) is an exponentially low probability event. The results in Section 3 also demonstrate a large amount of switching in terms of which trained variable is closest to each ground truth variable – which further indicates the way optimization is improved by overparameterization is more complicated.

More recently, Li et al. (2018) explored matrix completion and Xu et al. (2018) mixtures of *two* Gaussians. In Li et al. (2018), the authors consider fitting a *full-rank* matrix to the partially observed matrix, yet prove gradient descent finds the correct, low-rank matrix. This setting is substantially simpler than ours: the authors can leverage linearity in the analysis (in a suitable sense, matrix completion can be viewed as a noisy version of matrix factorization, for which gradient descent has a relatively simple behavior). Noisy-OR has substantive nonlinearities (in fact, this makes analyzing even non-gradient descent algorithms involved (Arora et al., 2017)), and sparse coding is complicated by sparsity in the latent variables.

In Xu et al. (2018), the authors prove that when fitting a symmetric, equal-weight, two-component mixture, treating the weights as variables helps EM avoid local minima. (This flies in contrast to the intuition that knowing that the weights are equal, one should incorporate this information directly into the algorithm.) This setting is also much simpler than ours: their analysis does not even generalize to non-symmetric mixtures and relies on the fact that the updates have a simple closed-form.

# 6 DISCUSSION

The goal of this work was to exhibit the first controlled and thorough study of the benefits of overparameterization in unsupervised learning settings, more concretely noisy-OR networks and sparse coding. The results show that overparameterization is beneficial and impervious to a variety of changes in the settings of the learning algorithm. We believe that our empirical study provides strong motivation for a program of theoretical research to understand the limits of when and how gradient-based optimization of the likelihood (or its approximations) can succeed in parameter recovery for unsupervised learning of latent variable models.

We note that overparameterization is a common strategy in practice – though it's usually treated as a recipe to develop more fine-grained latent variables (e.g., more specific topics in a topic model). In our paper, in contrast, we precisely document the extent to which it can aid optimization and enable recovering the ground-truth model.

We also note that our filtering step, while appealingly simple, will likely be insufficient in more complicated scenarios – e.g. when the activation priors have a more heavy-tailed distribution. In such settings, more complicated variable selection procedures would have to be devised, tailored to the distribution of the priors.

As demonstrated in Section 3, the choice of variational distribution has impact on the performance which cannot be offset by more overparameterization. In fact, when using the weaker variational approximation in Šingliar & Hauskrecht (2006) (which introduced some of the datasets we use), we were not able to recover all sources, regardless of the level of overparameterization. This delicate interplay between the power of the variational family and the level of overparameterization demands more study.

Inextricably linked to our study is precise understanding of the effects of architecture – especially so with the deluge of different varieties of (deep) generative models. We leave the task of designing controlled experiments for more complicated settings for future work.

Finally, the work of Zhang et al. (2016) on overparameterization in supervised settings considered a *data-poor* regime: where the number of parameters in the neural networks is comparable or larger than the training set. We did not explore such extreme levels of overparameterization in our work.

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

## A    APPENDIX

## A    LEARNING ALGORITHMS

We give here pseudocode for the learning algorithms, as well as details that were not provided in the main document.

### A.1    NOISY-OR NETWORKS WITH RECOGNITION NETWORK

Recall the evidence lower bound (ELBO):

$$\log p(x; \theta) \geq \mathbb{E}_{q(\cdot|x;\phi)} \left[ \log p(x, h; \theta) - \log q(h|x; \phi) \right] = \mathcal{L}(x, \theta, \phi)$$

One can derive the following gradients (see (Mnih & Gregor, 2014)):

$$\nabla_\theta \mathcal{L}(x, \theta, \phi) = \mathbb{E}_{q(\cdot|x;\phi)} [\nabla_\theta \log p(x, h; \theta)]$$

$$\nabla_\phi \mathcal{L}(x, \theta, \phi) = \mathbb{E}_{q(\cdot|x;\phi)} [(\log p(x, h; \theta) - \log q(h|x; \phi)) \times \nabla_\phi \log q(h|x; \phi)]$$

Then, we maximize the lower bound by taking gradient steps w.r.t. $\theta$ and $\phi$. To estimate the gradients, we average the quantities inside the expectations over multiple samples from $q(\cdot|x; \phi)$.

See Algorithm 1 for an update step of the algorithm, without variance normalization and input-dependent signal centering.

The experiments use mini-batch size 20 (unless specified otherwise). $\theta$ and $\phi$ are optimized using Adam.

---

**Algorithm 1** Update step for learning noisy-OR networks with a recognition network. $p(\cdot, \cdot; \theta)$ is the current noisy-OR network model (with parameters $\theta$), $q(\cdot|\cdot; \phi)$ is the current recognition network model (with parameters $\phi$), and $x$ is a batch of 20 samples.

---

**function** Update$(p, q, x)$
    **for** $i \leftarrow 1$ to 20 **do**
        $h^{(i)} \leftarrow \text{sample}(q(\cdot|x^{(i)}; \phi))$
        $r^{(i)} \leftarrow \log p(h^{(i)}, x^{(i)}; \theta) - \log q(h^{(i)}|x^{(i)}; \phi)$
    **end for**
    $\Delta\phi \leftarrow \Delta\phi + \eta \cdot \frac{1}{20} \sum_{i=1}^{20} r^{(i)} \cdot \nabla_\phi \log q(h^{(i)}|x^{(i)}; \phi)$
    $\Delta\theta \leftarrow \Delta\theta + \eta \cdot \frac{1}{20} \sum_{i=1}^{20} \nabla_\theta \log p(h^{(i)}, x^{(i)}; \theta)$
**end function**

---

#### A.1.1    VARIANCE NORMALIZATION AND INPUT-DEPENDENT SIGNAL CENTERING

We use variance normalization and input-dependent signal centering to improve the estimation of $\nabla_\theta \mathcal{L}(x, \theta, \phi)$, as in (Mnih & Gregor, 2014).

The goal of both techniques is to reduce the variance in the estimation of $\nabla_\phi \mathcal{L}(x, \theta, \phi)$. They are based on the observation that (see (Mnih & Gregor, 2014)):

$$\nabla_\phi \mathcal{L}(x, \theta, \phi) = \mathbb{E}_{q(\cdot|x;\phi)} [(\log p(x, h; \theta) - \log q(h|x; \phi)) \times \nabla_\phi \log q(h|x; \phi)]$$
$$= \mathbb{E}_{q(\cdot|x;\phi)} [(\log p(x, h; \theta) - \log q(h|x; \phi) - c) \times \nabla_\phi \log q(h|x; \phi)]$$

where $c$ does not depend on $h$. Therefore, it is possible to reduce the variance in the estimator by using some $c$ close to $\log p(x, h; \theta) - \log q(h|x; \phi)$.

Variance normalization keeps running averages of the mean and variance of $\log p(x, h; \theta) - \log q(h|x; \phi)$. Let $c$ be the average mean and $v$ be the average variance. Then variance normalization transforms $\log p(x, h; \theta) - \log q(h|x; \phi)$ into $\frac{\log p(x, h; \theta) - \log q(h|x; \phi) - c}{\max(1, \sqrt{v})}$.

Input-dependent signal centering keeps an input-dependent function $c(x)$ that approximates the normalized value of $\log p(x, h; \theta) - \log q(h|x; \phi)$. We model $c(x)$ as a two-layer neural network

with 100 hidden nodes in the second layer and $\tanh$ activation functions. We train $c(x)$ to minimize

$$\mathbb{E}_{q(\cdot|x;\phi)}\left[\left(\frac{\log p(x,h;\theta) - \log q(h|x;\phi) - c}{\max(1,\sqrt{v})} - c(x)\right)^2\right]$$

and optimize it using SGD.

Therefore, our estimator of $\nabla_\phi \mathcal{L}(x,\theta,\phi)$ is obtained as:

$$\nabla_\phi \mathcal{L}(x,\theta,\phi) \approx \hat{\mathbb{E}}_{q(\cdot|x;\phi)}\left[\left(\frac{\log p(x,h;\theta) - \log q(h|x;\phi) - c}{\max(1,\sqrt{v})} - c(x)\right) \times \nabla_\phi \log q(h|x;\phi)\right]$$

## A.2 NOISY-OR NETWORKS WITH MEAN-FIELD VARIATIONAL POSTERIOR

In the mean-field algorithm, we use the variational posterior $q(h) = \prod_{i=1}^m q_i(h_i)$. That is, the latent variables are modeled as independent Bernoulli.

For each data point, we optimize $q$ from scratch (unlike the case of the recognition network variational posterior, which is "global"), and then we make a gradient update to the generative model.

To optimize the variational posterior $q$ we use coordinate ascent, according to:

$$q_i(h_i) \propto \exp\{\mathbb{E}_{[m]\setminus\{i\}}[\log p(h_i, h_{[m]\setminus\{i\}}, x)]\}$$

where the expectation is over $h_{[m]\setminus\{i\}}$.

See Algorithm 2 for an update step of the algorithm. We use 5 iterations of coordinate ascent, and we use 20 samples to estimate expectations.

---

**Algorithm 2** Update step for learning noisy-OR networks with mean-field variational posterior. $p(\cdot, \cdot; \theta)$ is the current noisy-OR network model (with parameters $\theta$), and $x$ is a sample.

---

**function** Update$(p, x)$
  $q \leftarrow$ independent_Bernoulli_variables_distribution$(m)$
  **for** $iter \leftarrow 1$ to $5$ **do**
    **for** $k \leftarrow 1$ to $m$ **do**
      $E = [0, 0]$
      **for** $t \leftarrow 1$ to $20$ **do**
        $h_k = 0$
        $h_{[m]\setminus\{k\}} \leftarrow$ sample$(q(\cdot))$
        $E[0] \leftarrow E[0] + \frac{1}{20}\log p(h, x; \theta)$
        $E[1] \leftarrow E[1] + \frac{1}{20}\log p(h + 1_k, x; \theta)$
      **end for**
      $q_k(\cdot) \propto \exp\{E[\cdot]\}$
    **end for**
  **end for**
  **for** $i \leftarrow 1$ to $20$ **do**
    $h \leftarrow$ sample$(q(\cdot))$
    $\Delta\theta \leftarrow \Delta\theta + \eta \cdot \frac{1}{20} \cdot \nabla_\theta \log p(h, x; \theta)$
  **end for**
**end function**

---

## A.3 SPARSE CODING

See Algorithm 3 for an update step of the algorithm. We use batch size 20 in the learning algorithm in all experiments.

---

**Algorithm 3** Alternating minimization algorithm update for sparse coding. $A$ is the current matrix, and $x$ is a batch of 20 samples.

---

1: **function** Update($A, x$)
2:     **for** $i \leftarrow 1$ to 20 **do**
3:         $h^{(i)} \leftarrow \max(0, A^\dagger x^{(i)} - \alpha)$
4:     **end for**
5:     $A \leftarrow A - \eta \cdot \frac{1}{20} \sum_{i=1}^{20} [(Ah^{(i)} - x^{(i)})h^{(i)T}]$
6:     normalize columns of $A$
7: **end function**

---

## B   IMG DATA SET PROPERTIES

In addition to being easy to visualize, the noisy-OR network model of the IMG data set has properties that ensure it is not "too easy" to learn. Specifically, $5$ out of the $8$ latent variable do not have "anchor" observed variables (i.e. observed variables for which a single failure probability is different from $1.0$). Such anchor observed variables are an ingredient of most known provable algorithms for learning noisy-OR networks. More technically, the model requires a subtraction step in the quartet learning approach of (Jernite et al., 2013).

## C   PLNT DATA SET CONSTRUCTION

We learn the PLNT model from the UCI plants data set (Lichman et al., 2013), where each data point represents a plant that grows in North America and the 70 binary features indicate in which states and territories of North America it is found. The data set contains 34,781 data points.

To learn the data set, we use the learning algorithm described in Section A.1, with 20 latent variables. We remove all learned latent variables with prior probability less than $0.01$. Furthermore, we transform all failure probabilities greater than $0.50$ into $1.00$. This transformation is necessary to obtain sparse connections between the latent variables and the observed variables; without it, every latent variable is connected to almost every observed variable, which makes learning difficult. The resulting noisy-OR network model has $8$ latent variables. Each latent variable has a prior probability between $0.05$ and $0.20$. By construction, each failure probability different from $1.00$ is between $0.00$ and $0.50$.

Figure 4 shows a representation of the latent variables learned in the PLNT data set. As observed, the latent variables correspond to neighboring regions in North America.

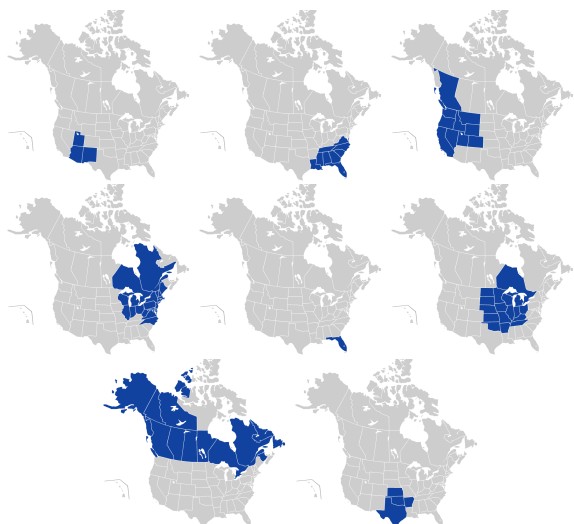

Figure 4: Latent variable configuration of the PLNT noisy-OR network model. Each map represents a latent variable. The regions in blue represent the observed variables for which the failure probability is not $1.00$. The fifth latent variable, which seems to contain only Florida, also contains Puerto Rico and Virgin Islands (not shown on map).

# D    DETAILED EXPERIMENT CONFIGURATION

Below we give more details on the configuration of the experiments.

## D.1    INITIALIZATION

In all noisy-OR network experiments, the noisy-OR network model is initialized by sampling each prior probability $\pi_i$, each failure probability $f_{ji}$, and each complement of noise probability $1 - l_j$ as follows: sample $z \sim \mathrm{uniform}[2.0, 4.0]$, and then set the parameter to $\mathrm{sigmoid}(z)$. Note that $\mathrm{sigmoid}(z)$ is roughly between $0.88$ and $0.98$, so the noisy-OR network is biased toward having large prior probabilities, large failure probabilities, and small noise probabilities. We found that this biased initialization improves results over one centered around $0.5$.

In the recognition network noisy-OR experiments, we initialize the recognition network to have all weight parameters and bias parameters uniform in $[-0.1, 0.1]$.

In the mean-field network noisy-OR experiments, we initialize the mean-field Bernoulli random variables to have parameters uniform in $[0.2, 0.8]$.

In the sparse coding experiments, we initialize the matrix by sampling each entry from a standard Gaussian, and then normalizing the columns such as to have unit $l_2$ norm.

## D.2    HYPERPARAMETERS

We generally tune hyperparameters within factors of $10$ for each experiment configuration. Due to time constraints, for each choice of hyperparameters we only test it on $10$ runs with random initializations of the algorithm, and then choosing the best performing hyperparameters for the large-scale experiments.

In the recognition network noisy-OR experiments, we tune the step size for the noisy-OR network model parameters and the step size for the input-dependent signal centering neural network. The step size for the recognition network model parameters is the same as the one for the noisy-OR network model parameters (tuning it independently did not seem to change the results significantly). The variance reduction technique requires a rate for the running estimates; we set this to $0.8$.

In the mean-field noisy-OR experiments, we only tune the step size for the noisy-OR network model parameters. For the coordinate ascent method used to optimize the mean-field parameters, we use 5 iterations of coordinate ascent (i.e. 5 iterations through all coordinates), and in each iteration we estimate expectations with 20 samples.

In the sparse coding experiments, we tune the step size for the updates to the matrix and the $\alpha$ variable.

### D.3 NUMBER OF EPOCHS

In all experiments, we use a large enough fixed number of epochs such that the log-likelihood / error measurement does not improve at the end of the optimization in all or nearly-all the runs. However, to avoid overfitting, we save the model parameters at regular intervals in the optimization process, and report the results from the timestep that achieved the best held-out log-likelihood / error measurement (i.e. we perform post-hoc "early stopping").

## E    TABLES OF RESULTS

Below we present detailed tables of results for the experiments.

Table 1 shows the noisy-OR network results with recognition network (partially included in the main document).

Table 2 shows the noisy-OR network results with recognition network and larger batch size 1000.

Table 3 shows the noisy-OR network results with mean-field variational posterior.

Table 4 shows the noisy-OR network results with recognition network on the misspecified data sets (IMG-FLIP and IMG-UNIF).

Table 5 shows the sparse coding results.

Table 1: Performance of the noisy-OR network learning algorithm with recognition network. Each row reports statistics for 500 runs of the algorithm with random initializations. The 95% confidence intervals are included. The first column denotes the number of latent variables used in learning, the second column the average number of ground truth latent variables recovered, the third column the percentage of runs with full ground truth recovery, and the fourth column the average held-out negative log-likelihood.

| LAT VARS | RECOV VARS | FULL RECOV (%) | NEGATIVE LL (NATS) |
|---|---|---|---|
| **IMG** | | | |
| 8 | $6.31 \pm 0.11$ | $29.6 \pm 4.0$ | $13.91 \pm 0.12$ |
| 16 | $7.62 \pm 0.06$ | $73.6 \pm 3.9$ | $12.65 \pm 0.05$ |
| 32 | $7.75 \pm 0.05$ | $79.6 \pm 3.5$ | $12.57 \pm 0.03$ |
| 64 | $7.73 \pm 0.05$ | $77.0 \pm 3.7$ | $12.56 \pm 0.03$ |
| 128 | $7.75 \pm 0.04$ | $75.6 \pm 3.8$ | $12.57 \pm 0.02$ |
| **PLNT** | | | |
| 8 | $4.71 \pm 0.12$ | $0.4 \pm 0.6$ | $12.36 \pm 0.07$ |
| 16 | $6.83 \pm 0.12$ | $45.0 \pm 4.4$ | $11.83 \pm 0.07$ |
| 32 | $6.57 \pm 0.11$ | $38.6 \pm 4.3$ | $12.23 \pm 0.08$ |
| 64 | $7.03 \pm 0.10$ | $55.0 \pm 4.4$ | $12.03 \pm 0.07$ |
| 128 | $7.58 \pm 0.08$ | $77.6 \pm 3.7$ | $11.96 \pm 0.05$ |
| **UNIF** | | | |
| 8 | $5.35 \pm 0.14$ | $12.6 \pm 2.9$ | $12.09 \pm 0.14$ |
| 16 | $7.78 \pm 0.05$ | $85.4 \pm 3.1$ | $10.10 \pm 0.03$ |
| 32 | $7.87 \pm 0.04$ | $88.2 \pm 2.8$ | $10.08 \pm 0.01$ |
| 64 | $7.91 \pm 0.04$ | $93.6 \pm 2.2$ | $10.08 \pm 0.01$ |
| 128 | $7.78 \pm 0.08$ | $91.2 \pm 2.5$ | $10.11 \pm 0.01$ |
| **CON8** | | | |
| 8 | $3.70 \pm 0.15$ | $1.2 \pm 1.0$ | $8.42 \pm 0.10$ |
| 16 | $5.77 \pm 0.15$ | $23.6 \pm 3.7$ | $7.15 \pm 0.08$ |
| 32 | $7.45 \pm 0.08$ | $71.6 \pm 4.0$ | $6.31 \pm 0.04$ |
| 64 | $7.68 \pm 0.06$ | $81.6 \pm 3.4$ | $6.38 \pm 0.03$ |
| 128 | $7.88 \pm 0.04$ | $92.8 \pm 2.3$ | $6.43 \pm 0.02$ |
| **CON24** | | | |
| 8 | $2.26 \pm 0.15$ | $0.4 \pm 0.6$ | $14.89 \pm 0.17$ |
| 16 | $4.90 \pm 0.21$ | $17.2 \pm 3.3$ | $11.17 \pm 0.13$ |
| 32 | $7.21 \pm 0.10$ | $53.8 \pm 4.4$ | $10.15 \pm 0.05$ |
| 64 | $7.60 \pm 0.06$ | $68.4 \pm 4.1$ | $11.75 \pm 0.14$ |
| 128 | $7.34 \pm 0.08$ | $52.2 \pm 4.4$ | $15.61 \pm 0.24$ |

Table 2: Performance of the noisy-OR network learning algorithm with recognition network and batch size 1000. Each row reports statistics for 500 runs of the algorithm with random initializations. The 95% confidence intervals are included. The first column denotes the number of latent variables used in learning, the second column the average number of ground truth latent variables recovered, the third column the percentage of runs with full ground truth recovery, and the fourth column the average held-out negative log-likelihood.

| LAT VARS | RECOV VARS | FULL RECOV (%) | NEGATIVE LL (NATS) |
|---|---|---|---|
| **IMG** | | | |
| 8 | $5.91 \pm 0.13$ | $24.2 \pm 3.8$ | $14.23 \pm 0.13$ |
| 16 | $7.17 \pm 0.10$ | $53.6 \pm 4.4$ | $13.01 \pm 0.10$ |
| 32 | $7.32 \pm 0.08$ | $58.6 \pm 4.3$ | $12.82 \pm 0.08$ |
| **PLNT** | | | |
| 8 | $4.35 \pm 0.12$ | $0.0 \pm 0.0$ | $13.74 \pm 0.05$ |
| 16 | $5.69 \pm 0.09$ | $5.0 \pm 1.9$ | $13.07 \pm 0.05$ |
| 32 | $6.18 \pm 0.08$ | $15.6 \pm 3.2$ | $12.76 \pm 0.06$ |
| **UNIF** | | | |
| 8 | $5.62 \pm 0.17$ | $26.6 \pm 3.9$ | $11.64 \pm 0.13$ |
| 16 | $6.20 \pm 0.19$ | $51.0 \pm 4.4$ | $10.35 \pm 0.05$ |
| 32 | $5.84 \pm 0.21$ | $49.2 \pm 4.4$ | $10.30 \pm 0.04$ |
| **CON8** | | | |
| 8 | $2.86 \pm 0.14$ | $0.0 \pm 0.0$ | $8.89 \pm 0.10$ |
| 16 | $4.31 \pm 0.16$ | $6.0 \pm 2.1$ | $7.98 \pm 0.10$ |
| 32 | $7.22 \pm 0.10$ | $66.2 \pm 4.2$ | $6.25 \pm 0.06$ |
| **CON24** | | | |
| 8 | $2.52 \pm 0.18$ | $2.0 \pm 1.2$ | $14.67 \pm 0.20$ |
| 16 | $5.67 \pm 0.24$ | $51.8 \pm 4.4$ | $11.63 \pm 0.24$ |
| 32 | $6.86 \pm 0.19$ | $74.4 \pm 3.8$ | $10.49 \pm 0.19$ |

Table 3: Performance of the noisy-OR network learning algorithm with mean-field variational posterior. Each row reports statistics for 500 runs of the algorithm with random initializations. The 95% confidence intervals are included. The first column denotes the number of latent variables used in learning, the second column the average number of ground truth latent variables recovered, the third column the percentage of runs with full ground truth recovery, and the fourth column the average held-out negative log-likelihood.

| LAT VARS | RECOV VARS | FULL RECOV (%) | NEGATIVE LL (NATS) |
|---|---|---|---|
| **IMG** | | | |
| 8 | $6.72 \pm 0.10$ | $41.8 \pm 4.3$ | $13.50 \pm 0.11$ |
| 16 | $7.52 \pm 0.07$ | $66.4 \pm 4.1$ | $12.58 \pm 0.06$ |
| 32 | $7.23 \pm 0.09$ | $56.2 \pm 4.4$ | $12.80 \pm 0.08$ |
| **PLNT** | | | |
| 8 | $4.35 \pm 0.12$ | $0.0 \pm 0.0$ | $13.74 \pm 0.05$ |
| 16 | $5.69 \pm 0.09$ | $5.0 \pm 1.9$ | $13.07 \pm 0.05$ |
| 32 | $6.18 \pm 0.08$ | $15.6 \pm 3.2$ | $12.76 \pm 0.06$ |
| **UNIF** | | | |
| 8 | $5.62 \pm 0.17$ | $26.6 \pm 3.9$ | $11.64 \pm 0.13$ |
| 16 | $6.20 \pm 0.19$ | $51.0 \pm 4.4$ | $10.35 \pm 0.05$ |
| 32 | $5.84 \pm 0.21$ | $49.2 \pm 4.4$ | $10.30 \pm 0.04$ |
| **CON8** | | | |
| 8 | $2.86 \pm 0.14$ | $0.0 \pm 0.0$ | $8.89 \pm 0.10$ |
| 16 | $4.31 \pm 0.16$ | $6.0 \pm 2.1$ | $7.98 \pm 0.10$ |
| 32 | $7.22 \pm 0.10$ | $66.2 \pm 4.2$ | $6.25 \pm 0.06$ |
| **CON24** | | | |
| 8 | $2.52 \pm 0.18$ | $2.0 \pm 1.2$ | $14.67 \pm 0.20$ |
| 16 | $5.67 \pm 0.24$ | $51.8 \pm 4.4$ | $11.63 \pm 0.24$ |
| 32 | $6.86 \pm 0.19$ | $74.4 \pm 3.8$ | $10.49 \pm 0.19$ |

Table 4: Performance of the noisy-OR network learning algorithm with recognition network on the misspecified data sets (IMG-FLIP and IMG-UNIF). Each row reports statistics for 500 runs of the algorithm with random initializations. The 95% confidence intervals are included. The first column denotes the number of latent variables used in learning, the second column the average number of ground truth latent variables recovered, the third column the percentage of runs with full ground truth recovery, and the fourth column the average held-out negative log-likelihood.

| LAT VARS | RECOV VARS | FULL RECOV (%) | NEGATIVE LL (NATS) |
|---|---|---|---|
| **IMG-FLIP** | | | |
| 8 | $4.40 \pm 0.10$ | $0.2 \pm 0.4$ | $17.55 \pm 0.10$ |
| 9 | $5.59 \pm 0.13$ | $12.2 \pm 2.9$ | $16.49 \pm 0.11$ |
| 10 | $5.99 \pm 0.13$ | $18.2 \pm 3.4$ | $15.98 \pm 0.10$ |
| 16 | $6.88 \pm 0.09$ | $27.0 \pm 3.9$ | $15.10 \pm 0.06$ |
| **IMG-UNIF** | | | |
| 8 | $4.95 \pm 0.12$ | $0.0 \pm 0.0$ | $21.51 \pm 0.08$ |
| 9 | $5.44 \pm 0.12$ | $6.0 \pm 2.1$ | $20.77 \pm 0.09$ |
| 16 | $7.27 \pm 0.09$ | $59.0 \pm 4.3$ | $18.05 \pm 0.08$ |
| 32 | $7.76 \pm 0.05$ | $80.0 \pm 3.5$ | $16.41 \pm 0.06$ |

Table 5: Performance of the sparse coding learning algorithm. Each row reports statistics for 500 runs of the algorithm with random initializations. The 95% confidence intervals are included. The first column denotes the number of latent variables used in learning, the second column the average number of ground truth columns recovered, the third column the percentage of runs with full ground truth recovery, and the fourth column the average error.

| COLS | RECOV COLS | FULL RECOV (%) | ERROR |
|---|---|---|---|
| $\gamma = 5°$ | | | |
| 24 | $8.71 \pm 0.18$ | $0.0 \pm 0.0$ | $9.32 \pm 0.18$ |
| 48 | $14.54 \pm 0.37$ | $1.4 \pm 1.0$ | $3.11 \pm 0.16$ |
| $\gamma = 10°$ | | | |
| 24 | $19.77 \pm 0.29$ | $17.2 \pm 3.3$ | $2.28 \pm 0.15$ |
| 48 | $23.84 \pm 0.05$ | $88.4 \pm 2.8$ | $0.04 \pm 0.02$ |

## F STATE OF THE OPTIMIZATION PROCESS

Figures 5 and 6 show more steps of the optimization process on the successful run with 16 latent variables mentioned in Figure 3.

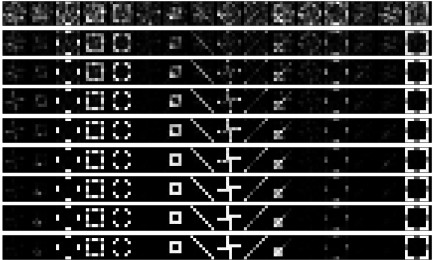

Figure 5: Latent variables on a successful run of the noisy-OR network learning algorithm on the IMG data set with 16 latent variables. Shown is the state of the latent variables after epochs $1/9$, $2/9$, $3/9$, $4/9$, $5/9$, $6/9$, $8/9$, $8/9$, and 1.



Figure 6: Latent variables on a successful run of the noisy-OR network learning algorithm on the IMG data set with 16 latent variables. Shown is the state of the latent variables after epochs 1, 2, 3, 5, 10, 20, 30, 50, and 100.

## G RECOVERED LATENT VARIABLES ON MISSPECIFIED DATA SETS

Figure 7 shows successful recoveries for the IMG-FLIP and IMG-UNIF data sets. As observed, some of the extra latent variables are used to model some of the noise due to misspecification.

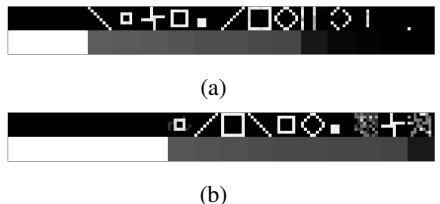

(a)

(b)

Figure 7: Latent variables recovered in successful runs (i.e. they recover the IMG latent variables) on the mismatched data sets. Below each $8 \times 8$ images corresponding to a latent variable, there is a color corresponding to its prior (whiter means prior closer to $1.0$). (a) Successful run with 16 latent variables on the IMG-FLIP data set. (b) Successful run with 16 latent variables on the IMG-UNIF data set.

## H  DUPLICATE LATENT VARIABLES ARE NOT EQUIVALENT TO A SINGLE LATENT VARIABLE

We give an example of a model with two latent variables that has identical failure probabilities and is not equivalent to a model with one latent variable that has the same failure probabilities (and possibly different priors).

Consider a noisy-OR network model with two latent variables $h_1, h_2$ and two observed variables $x_1, x_2$. Let $\pi_1 = \pi_2 = 0.25$ (prior probabilities), $f_{11} = f_{12} = f_{21} = f_{22} = 0.1$ (failure probabilities), and $l_1 = l_2 = 0.0$ (noise probabilities). Then the negative moments are $\mathbb{P}(x_1 = 0) = 0.600625$, $\mathbb{P}(x_2 = 0) = 0.600625$, and $\mathbb{P}(x_1 = 0, x_2 = 0) = 0.56625625$.

Consider now a noisy-OR network model with one latent variable $h'_1$ and two observed variables $x'_1, x'_2$. Let $f'_{11} = f'_{21} = 0.1$ and $l'_1 = l'_2 = 0.0$. Then, to match the first-order negative moments $\mathbb{P}(x'_1 = 0) = \mathbb{P}(x_1 = 0)$ and $\mathbb{P}(x'_2 = 0) = \mathbb{P}(x_2 = 0)$, we need $\pi'_1 = 0.44375$ (prior probability). But then this gives $\mathbb{P}(x'_1 = 0, x'_2 = 0) = 0.5606875$, which does not match $\mathbb{P}(x_1 = 0, x_2 = 0)$. Therefore, there exists no noisy-OR model with one latent variable and identical failure and noise probabilities that is equivalent to the noisy-OR model with two latent variables.

