# OpenReview forum: "Benefits of Overparameterization in Single-Layer Latent Variable Generative Models"
_ICLR.cc/2020/Conference — Reject_

### Official Review · AnonReviewer2 · 2019-10-23
**Official Blind Review #2**

**Rating:** 3

**Review:**

This paper investigates benefit of over-parameterization for latent variable generative model while existing researches typically focus on supervised learning settings. It is experimentally shown that the over-parameterization helps to obtain better optimization, but too much over-parameterization gives performance deterioration. In the numerical experiments, the effect of over-parameterization is investigated from several aspects.

The motivation of this paper is interesting. The writing of the paper is clear, and I could follow the contents easily.

On the other hand, I have the following concerns on the significance of the paper.
- All datasets investigated in this paper are rather small. If there were thorough investigations on more modern deep generative models, then the paper would be stronger. For example, the latent variable model is recently well discussed in the context of disentanglement representation. The generative models to obtain disentanglement representation could be investigated in the frame-work of this paper.
- This is an empirical study, but if there was theory to support the empirical observations, then the paper was more convincing. The problem itself is just a sparse coding problem. Hence, I think what investigated in this paper can be discussed by relating sparse coding theories. However, there is no theoretical justification on the experimental results.
- Summarizing the above arguments, the insight obtained in this paper is a bit weak. More ablation study and more experiments on general models will clarify what is going on in the over-parameterized model for latent generative models.

**Experience Assessment:**

I have read many papers in this area.

**Review Assessment: Checking Correctness Of Derivations And Theory:**

N/A

**Review Assessment: Checking Correctness Of Experiments:**

I carefully checked the experiments.

**Review Assessment: Thoroughness In Paper Reading:**

I read the paper at least twice and used my best judgement in assessing the paper.

---

> ### Author Response · Authors · 2019-11-15
> **Response to Review #2**
>
> Thank you for your comments!
>
> We agree that a study of deep generative models would be very interesting, but we see this work as a necessary prerequisite. By focusing on the simplest linear (sparse coding) and non-linear (noisy-OR) models in which the beneficial effect of overparameterization manifests, it allowed us to determine precisely how variations in the parameters of the ground-truth model and algorithms affect it.
>
> There are several key challenges in moving to deep generative models:
> 1. Even basic questions of identifiability are not well understood. Specifically, for parameter recovery to even make sense, the underlying generative process (i.e., the parameters for the p(z,x) distribution) has to be identifiable from the marginal distribution p(x). Results in this vein are known for sparse coding and noisy-OR networks, but not for deep generative models; note that the recent papers on learning disentangled representations do not include synthetic experiments where data is drawn from a deep generative model and the resulting model is shown to be “recovered”.
> 2. Depending on the architecture, there are many different ways to overparametrize a deeper model. (One could overparametrize in terms of depth, width, in some structurally constrained way, etc.)
> 3. Designing filtering/variable extraction steps to recover the ground truth variables is entirely unclear. It’s likely that the outcome of the experiment will vary significantly depending on the implementation of this step, and the set of potential choices is vast.
>
> On theory: while it would be great to have theory accompanying our empirical observations, we note that theoretical analysis for our settings is likely to be very non-trivial given our current understanding of the optimization for these latent-variable models. For noisy-OR networks, the currently known algorithms with provable guarantees use tensor-based techniques and are very different from the gradient-descent algorithm used by us (e.g. see [1] and [2]). For sparse coding, the currently known results about gradient-descent like algorithms assume incoherence of the ground truth matrix, as well as the *iterates* of the algorithm (e.g. see [3] and [4]). It is clear that the iterates will not be incoherent in our setup due to the existence of near-duplicates -- so such techniques seem difficult to generalize.
>
> [1] Jernite, Yacine, Yonatan Halpern, and David Sontag. "Discovering hidden variables in noisy-or networks using quartet tests." In Advances in Neural Information Processing Systems, pp. 2355-2363. 2013.
>
> [2] Arora, Sanjeev, Rong Ge, Tengyu Ma, and Andrej Risteski. "Provable learning of noisy-or networks." In Proceedings of the 49th Annual ACM SIGACT Symposium on Theory of Computing, pp. 1057-1066. ACM, 2017.
>
> [3] Arora, Sanjeev, Rong Ge, Tengyu Ma, and Ankur Moitra. "Simple, efficient, and neural algorithms for sparse coding." (2015).
>
> [4] Chatterji, Niladri, and Peter L. Bartlett. "Alternating minimization for dictionary learning: Local Convergence Guarantees."  arXiv:1711.03634

---

### Official Review · AnonReviewer3 · 2019-10-24
**Official Blind Review #3**

**Rating:** 6

**Review:**

The paper “aims to be a controlled empirical study making precise the benefits of overparameterization in unsupervised learning settings. ” The author’s empirical study is comprehensive, and to my knowledge the most detailed published work on this to date. Specifically, the authors empirically study
- the ability of networks to recover latent variables
- the effects of extreme overparameterization
- the effects the training method (e.g. batch size)
- latent variable stability over the course of training

In line with the findings for supervised settings, the authors find that overparameterization is often beneficial, and that overfitting is a surprisingly small issue. This is an interesting and useful observation, particularly since it at first sight appears to be in disagreement with some earlier work (the authors suggest explanations for the differing observations).

As the authors point out (and I agree), the paper constitutes a compelling reason for theoretical research on the interplay between overparameterization and parameter recovery in latent variable neural networks trained with gradient descent methods.

The authors perform studies on a range of different real-world and synthetic datasets.

The paper is well-written, well-structured, and easy to follow. Relevant literature has been cited. The appendices contain a wealth of details that will make this work reproducible.

Decision: weak accept. The paper contains some new insights, but its contributions are not quite as substantial (e.g. lack of precise mathematical statements) or surprising as those in stronger ICLR papers.

A small gripe: the authors promise “ a controlled empirical study making precise the benefits of overparameterization in unsupervised learning settings”. I would argue that “making precise” is too strong for what the paper actually delivers. I suggest rewording this.

**Experience Assessment:**

I have read many papers in this area.

**Review Assessment: Checking Correctness Of Derivations And Theory:**

I carefully checked the derivations and theory.

**Review Assessment: Checking Correctness Of Experiments:**

I assessed the sensibility of the experiments.

**Review Assessment: Thoroughness In Paper Reading:**

I read the paper thoroughly.

---

> ### Author Response · Authors · 2019-11-15
> **Response to Review #3**
>
> Thank you for your comments. We reworded the “making precise” sentence to now read, “a controlled empirical study that measures and disentangles the benefits of overparameterization in unsupervised learning settings.”
>
> About the lack of precise mathematical statements: while it would be great to have theory accompanying our empirical observations, we note that theoretical analysis for our settings is likely to be very non-trivial given our current understanding of the optimization for these latent-variable models. For noisy-OR networks, the currently known algorithms with provable guarantees use tensor-based techniques and are very different from the gradient-descent algorithm used by us (e.g. see [1] and [2]). For sparse coding, the currently known results about gradient-descent like algorithms assume incoherence of the ground truth matrix, as well as the *iterates* of the algorithm (e.g. see [3] and [4]). It is clear that the iterates will not be incoherent in our setup due to the existence of near-duplicates -- so such techniques seem difficult to generalize.
>
> [1] Jernite, Yacine, Yonatan Halpern, and David Sontag. "Discovering hidden variables in noisy-or networks using quartet tests." In Advances in Neural Information Processing Systems, pp. 2355-2363. 2013.
>
> [2] Arora, Sanjeev, Rong Ge, Tengyu Ma, and Andrej Risteski. "Provable learning of noisy-or networks." In Proceedings of the 49th Annual ACM SIGACT Symposium on Theory of Computing, pp. 1057-1066. ACM, 2017.
>
> [3] Arora, Sanjeev, Rong Ge, Tengyu Ma, and Ankur Moitra. "Simple, efficient, and neural algorithms for sparse coding." (2015).
>
> [4] Chatterji, Niladri, and Peter L. Bartlett. "Alternating minimization for dictionary learning: Local Convergence Guarantees."  arXiv:1711.03634

---

### Official Review · AnonReviewer1 · 2019-10-26
**Official Blind Review #1**

**Rating:** 3

**Review:**

This paper performs empirical study on the influence of overparameterization to generalization performance of noisy-or networks and sparse coding, and points out overparameterization is indeed beneficial. I find the paper has some drawbacks.

1. Overparameterization is better than underparamterization and exact parameterization is not surprising. The question is how much do we need to overparameterize. As the number of parameters goes to infinity, the model can eventually remember all the training data, and has poor generalization. The real interesting question to ask is how to use an excessive amount of parameters, yet still avoid overfitting.

2. The discussed models are too simple. I am expecting some theoretical analysis for tasks simple as noisy-or and sparse coding, or some experiments for more complicated (deep) models need to be done, to make the paper more solid.

Update
=====

Thank the authors for the response. The authors do address my comment #1. I agree that overparameterization improves recovery is a new finding. However, I still think the "information gain" of this paper is somewhat thin. There could be at least some intuitions on why overparameterization helps noisy-or models. I think the analysis can be more in-depth to make this paper more interesting.

I would like to raise my score a bit to a "neutral" score, but given the current scoring system I'll just keep my score.

**Experience Assessment:**

I do not know much about this area.

**Review Assessment: Checking Correctness Of Derivations And Theory:**

I assessed the sensibility of the derivations and theory.

**Review Assessment: Checking Correctness Of Experiments:**

I assessed the sensibility of the experiments.

**Review Assessment: Thoroughness In Paper Reading:**

I read the paper at least twice and used my best judgement in assessing the paper.

---

> ### Author Response · Authors · 2019-11-15
> **Response to Review #1**
>
> Thank you for your comments!
>
> Regarding (1), the rationale of the reviewer applies to the *likelihood* of the model we fit: the training-set likelihood should improve, and there is a potential that the test likelihood will drop (i.e. memorization happens). Our paper *does not* focus on this -- it focuses on *parameter recovery* -- here, it is entirely unclear why overparametrization should help, as there can be potentially many overparametrized models with equally good likelihood, but no relation to the ground truth parameters of the model whatsoever. This isn’t an issue of memorization -- rather in the presence of overparametrization, the model could in principle be un-identifiable (i.e. multiple sets of parameters give rise to the same distribution), so there is no a-priori reason why the optimization should prefer the ground truth parameters.
>
> Moreover, in our noisy-OR experiments, 128 latent variables are already 16 times more than the true number of latent variables, and at this level of overparameterization the performance is still much better than without overparameterization. Hence, we show that if there is a “critical” amount of overparametrization at which performance starts to suffer, it may be quite large.
>
> Regarding (2): while it would be great to have theory accompanying our empirical observations, we note that theoretical analysis for our settings is likely to be very non-trivial given our current understanding of the optimization for these latent-variable models. For noisy-OR networks, the currently known algorithms with provable guarantees use tensor-based techniques and are very different from the gradient-descent algorithm used by us (e.g. see [1] and [2]). For sparse coding, the currently known results about gradient-descent like algorithms assume incoherence of the ground truth matrix, as well as the *iterates* of the algorithm (e.g. see [3] and [4]). It is clear that the iterates will not be incoherent in our setup due to the existence of near-duplicates -- so such techniques seem difficult to generalize.
>
> We agree that a study of deep generative models would be very interesting, but we see this work as a necessary prerequisite. By focusing on the simplest linear (sparse coding) and non-linear (noisy-OR) models in which the beneficial effect of overparameterization manifests, it allowed us to determine precisely how variations in the parameters of the ground-truth model and algorithms affect it.
>
> There are several key challenges in moving to deep generative models:
> 1. Even basic questions of identifiability are not well understood. Specifically, for parameter recovery to even make sense, the underlying generative process (i.e., the parameters for the p(z,x) distribution) has to be identifiable from the marginal distribution p(x). Results in this vein are known for sparse coding and noisy-OR networks, but not for deep generative models; note that the recent papers on learning disentangled representations do not include synthetic experiments where data is drawn from a deep generative model and the resulting model is shown to be “recovered”.
> 2. Depending on the architecture, there are many different ways to overparametrize a deeper model. (One could overparametrize in terms of depth, width, in some structurally constrained way, etc.)
> 3. Designing filtering/variable extraction steps to recover the ground truth variables is entirely unclear. It’s likely that the outcome of the experiment will vary significantly depending on the implementation of this step, and the set of potential choices is vast.
>
> [1] Jernite, Yacine, Yonatan Halpern, and David Sontag. "Discovering hidden variables in noisy-or networks using quartet tests." In Advances in Neural Information Processing Systems, pp. 2355-2363. 2013.
>
> [2] Arora, Sanjeev, Rong Ge, Tengyu Ma, and Andrej Risteski. "Provable learning of noisy-or networks." In Proceedings of the 49th Annual ACM SIGACT Symposium on Theory of Computing, pp. 1057-1066. ACM, 2017.
>
> [3] Arora, Sanjeev, Rong Ge, Tengyu Ma, and Ankur Moitra. "Simple, efficient, and neural algorithms for sparse coding." (2015).
>
> [4] Chatterji, Niladri, and Peter L. Bartlett. "Alternating minimization for dictionary learning: Local Convergence Guarantees."  arXiv:1711.03634

---

### Decision · Program_Chairs · 2019-12-19

**Decision:**

Reject

**Comment:**

This paper studies over-parameterization for unsupervised learning. The paper does a series of empirical studies on this topic. Among other things the authors observe that larger models can increase the number latent variables recovered when fitting larger variational inference models. The reviewers raised some concern about the simplicity of the models studied and also lack of some theoretical justification. One reviewer also suggests that more experiments and ablation studies on more general models will further help clarify the role over-parameterized model for latent generative models. I agree with the reviewers that this paper is "compelling reason for theoretical research on the interplay between overparameterization and parameter recovery in latent variable neural networks trained with gradient descent methods". I disagree with the reviewers that theoretical study is required as I think a good empirical paper with clear conjectures is as important. I do agree with the reviewers however that for empirical paper I think the empirical studies would have to be a bit more thorough with more clear conjectures. In summary, I think the paper is nice and raises a lot of interesting questions but can be improved with more through studies and conjectures. I would have liked to have the paper accepted but based on the reviewer scores and other papers in my batch I can not recommend acceptance at this time. I strongly recommend the authors to revise and resubmit. I really think this is a nice paper and has a lot of potential and can have impact with appropriate revision.